# Modeling and Simulation as Tools to Increase the Protection of Critical Infrastructure and the Sustainability of the Provision of Essential Needs of Citizens

Tomáš Loveček *, Lenka Straková and Katarína Kampová

Faculty of Security Engineering, University of Zilina, Univerzitná 1, 010 26 Žilina, Slovakia; lenka.sivakova@fbi.uniza.sk (L.S.); katarina.kampova@fbi.uniza.sk (K.K.)
* Correspondence: tomas.lovecek@fbi.uniza.sk

**Abstract:** States and their cities are at the forefront of efforts to address many of today's transnational security challenges. States cannot fulfill their basic functions, which include caring for the all-round development of their territory and the needs of its inhabitants, without the existence of critical infrastructure, which can be damaged, destroyed, or disrupted by malicious behavior. The legislation of EU authorities presents methods for improving the crisis management cycle within critical infrastructure protection. However, these methods are not elaborated. Modeling and simulation using software tools enable more accurate decision-making by security managers during the process of designing and evaluating the physical protection systems of critical infrastructure. This article presents a new software solution to the intrusion of unauthorized persons and its potential mathematical extension. The main innovative benefit of this newly created software tool is the possibility of creating more sophisticated attack scenarios using various 2D maps. Mathematical extension model response scenarios are constructed for various types of intruders, allowing more accurate training of defenders, which leads to more effective resource usage. This combination of software and mathematical solutions should allow physical protection system designers to test various intrusion scenarios of critical infrastructure elements.

**Keywords:** critical infrastructure; physical protection system; modeling; game theory

## 1. Introduction

The 2030 Agenda for Sustainable Development was adopted by world leaders at the historic UN Summit in September 2015, and was officially enacted on 1 January 2016. Over the next 15 years, countries have committed to achieving the 17 Sustainable Development Goals (SDGs). Goal number 11 is to make cities and human settlements inclusive, safe, resilient, and sustainable. Cities are also at the forefront of efforts to address many of today's transnational security challenges, as well as climate change, natural disasters, and mass movements of people. Future cities need to be safe, sustainable, and resilient to disasters [1].

States, cities, and their infrastructure are constantly exposed to the negative effects of naturogenic and anthropogenic threats. The impact of climatic change is considered to be the most significant naturogenic global threat. However, together with the global threats, local threats also affect states, cities, and their critical infrastructure. Terrorism is considered to be the most significant long-term local threat. The most considerable impacts in the case of terrorist attacks are observed for critical infrastructure components. Ultimately, a local terrorist attack through a cascading effect can result in a natural disaster or even a change in climate (e.g., an attack on a nuclear facility).

States and cities would not be able to fulfill their basic functions, which include caring for the all-round development of its territory and the needs of its inhabitants, without the existence of critical infrastructure. Critical infrastructure around the globe serves humans

in multiple ways by catering to their essential needs for energy, water, food, healthcare, and transportation, to name but a few. These facilities are vital for the health, wellbeing, and economic growth of nations, and are pivotal contributors to sustainable development. Critical infrastructure can also directly or indirectly negatively affect the environment, economy, and society [2].

Critical infrastructure (CI) is an asset or system essential for the maintenance of vital social functions, health, safety, security, and the economic or social wellbeing of people. European critical infrastructure (ECI) is critical infrastructure in EU countries whose disruption or destruction would have a significant impact on at least two EU countries (e.g., electricity power plants or oil transmission pipelines) [3].

Critical infrastructure includes physical resources, services, information technology facilities, networks, and infrastructure assets that, if disrupted or destroyed, would have a serious impact on the critical societal functions, including the supply chain, health, safety, security, economic, or social wellbeing of people or of the functioning of the Community or its Member States [4].

Critical infrastructure can be damaged, destroyed, or disrupted by deliberate acts of terrorism, natural disasters, negligence, and accidents, or computer hacking, criminal activity, and malicious behavior. To save the lives and property of people at risk in the EU from terrorism, natural disasters, and accidents, any disruptions or manipulations of CI should, to the greatest extent possible, be brief, infrequent, manageable, geographically isolated, and minimally detrimental to the welfare of the Member States, their citizens, and the European Union. The recent terrorist attacks in Madrid and London have highlighted the risk of terrorist attacks against European infrastructure [5].

The major focus of this study was the protection of critical infrastructure from the deliberate actions of unauthorized persons intending to harm, destroy, or steal protected tangible and/or intangible assets of a specified object owned or managed by a physical or legal entity. This type of protection primarily follows binding legislation (e.g., laws, legislative decrees, regulations, and guidelines), national and/or international technical standards, or requirements of insurance companies or other third parties, such as parent companies or strategic customers [5].

In terms of the physical protection of critical infrastructure elements, the EU legislation follows a general directive approach. The Green Paper, issued by the Commission of the European Communities [5], presents some methods for improving prevention, protection, preparedness, and response within critical infrastructure protection in the EU. However, those methods are not specified in detail.

Software tools and mathematical models are widely used in the IT sector, such as modeling based on big data analysis [6]. However, for the security sector, it is necessary to model social phenomena with mathematical approaches, as shown in a study of Windows users' behavior [7]. It is even more difficult to model the influence of psychological aspects on social behavior [8]. Because of the complexity of social phenomena, the modeling of physical protection systems (PPSs) often narrows to the modeling of technical aspects of these systems, as previously reported [9,10].

Similarly, specific tools that may be used to mitigate the threat to critical infrastructure can incorporate technical protective measures to discourage, detect, verify, and eliminate the intruder (both mechanical and electronic) as well as enhance the operation of security services (e.g., intervention of security and armed forces) [11–16].

Such tools include:

- SAVI/ASSESS (Sandia National Laboratories, USA);
- Sprut (ISTA, Russia);
- Vega-2 (Eleron, Russia);
- Analyser SFZ (FRTK MFTI, Russia);
- SAPE (Korea Institute of Nuclear Non-proliferation and Control, South Korea);
- Assessment of Terrorist Attack in a Network of Objects (SATANO) version 1 (University of Zilina, Slovakia).

The Systematic Analysis of Vulnerability to Intrusion (SAVI) method combines Estimation of Adversary Sequence Interruption (EASI) and Adversary Sequence Diagram (ASD) methods, evaluates every possible path to the central zone from the perspective of the probability of interruption, and creates a list of the ten most vulnerable paths according to their possibilities of interruption. The Analytic System and Software for Evaluating Safeguards and Security (ASSESS) method is an extension of the SAVI method that contains additional modules for the analysis of intruder neutralization and of internal adversaries, as well as cooperation between internal and external adversaries. Systematic Analysis of Physical Protection Effectiveness (SAPE) is a tool for the evaluation of the effectiveness of security systems that follows SAVI and ASSESS methods but improves upon these significantly. This method uses a 2D model of the protected area instead of the ASD model and uses a new heuristic algorithm, which considerably extends sensitivity analysis. SPRUT is a software tool used for evaluating physical protection effectiveness in nuclear facilities. Software tools can enable the modeling of combat encounters between intruders and physical forces. Vega-2 is a software tool intended for the determination of physical protection system effectiveness for nuclear facilities in a specified security system structure and different adversary models (internal and external). A specific Russian tool is Analizator SFZ, which is intended for the calculation of the shortest time that a security system will be overcome [17,18].

SATANO is a software tool using an alternative approach based on mathematical models. The primary aim is to minimize the extent of the evaluator's personal influence and introduce a model based on the game theory, which allows the incorporation of psychological or societal aspects into the models of physical protection systems. However, this approach is used only minimally within Europe, primarily because current assessment tools do not have a value basis of probabilistic and time input parameters (e.g., probability of alarm system detection and breakthrough resistance times).

To overcome this issue, the University Science Park of the University of Zilina was constructed with a polygon laboratory of protection systems for critical infrastructure objects included inside. This workplace specialization, which is unique in all of Europe, allows the creation of polygons to measure values of probabilistic and time parameters for the above basic models assessing the protection system levels for critical infrastructure elements [18]. Polygons are a type of functional map of an object, an urban element and an element of critical infrastructure, which form the basis of the proposed system.. Each polygon requires input variables to make it functional.

To address the lack of input variable values for mathematical models, the PACITA project Methodology for assessing the physical protection of critical infrastructure elements against terrorist and other types of attacks (HOME/2010/CIPS/AG/044) was undertaken in 2012–2013 as part of the CIPS program and focused on penetration tests of perimeter protection fencing systems [19,20].

However, many other types of input parameters need to be determined to make software tools like SATANO effective. These parameters are estimated by experts if it is not possible to measure them. Therefore, another project, Minimization of the degree of subjectivity of estimates of experts in security practice using quantitative and qualitative methods within the VEGA project call (2017–2020), was undertaken at the University of Zilina. This project aims to more rigorously estimate single parameters or systems of estimated parameters.

Another issue is identifying and building strategies that can be used to affect the clash between intruders and defenders in physical protection systems.

For over 15 years, researchers at the University of Zilina have been addressing questions covering complex efficiency assessment and the effectiveness of PPSs for strategic state facilities against anthropogenic threats.

SATANO, with additional research on the generation of input parameters by measuring, estimation, etc., represents an effective and adaptable approach to critical infrastructure protection from the deliberate actions of unauthorized persons. The mathematical theory

and software solution that are being developed at the University of Zilina are presented in this article.

## 2. Materials and Methods

Security plans or equivalent measures should be established for every element of the critical infrastructure, for example, nuclear facilities, buildings, and premises for the storage and handling of classified information; financial institutions; essential services, and relevant technologies; as well as elements whose protection may not be addressed by legislation (e.g., line or junction constructions, road, air, water, and railway transport works, chemical plants, energy supply facilities, water structures, food-processing enterprises, industrial enterprises, and health facilities). The plans should identify the significant facilities, identify and assess risks, and select and prioritize countermeasures and procedures. Whereas the protection of elements that are of strategic importance is addressed individually by each nation in its legislation, security requirements for these elements should be evaluated based on a common minimum approach (Council of the European Union 2008).

Legislation can be divided into two groups. The first group is characterized as declarative and specifies duties of an employee, owner, or a lessee to protect their property, which is in their possession or management (e.g., civil code and labor code). The second group of legislation requires a specific approach to protect strategic objects, which, because of their activities, interfere with the state operation or affect the lives of a large number of people. Therefore, their activities can be incorporated into national or European critical infrastructure. According to the European Council Decision of 2007 [4], state-operated critical infrastructure predominantly includes the physical resources, services, information technology equipment, networks, and roads that, if damaged or destroyed, can negatively affect the lives of many people and seriously impair critical social functions, including the supply chain, health services, security, protection, the economic and social wellbeing of citizens, and the functioning of the European Union (EU) or its member states (Council of the European Union 2007).

There are three basic approaches to designing and assessing the level of a physical protection system (PPS) suitable for protecting strategic state assets [21].

Directive approach: The subject must adopt the precisely specified PPS irrespective of operational specificities and location [22]. This approach is used when the efficiency, reliability, or effectiveness of the designed protective measures cannot be verified, necessitating reliance on the professional experience and expertise of standards authors, binding legal regulations, or software application producers. This approach is represented by the Analysis of the Protection of Energy Networks' Crucial Objects against Terrorism (APENCOT) project; the Proposal of Security Standards project was implemented in 2008–2010 as part of a Chartered Institute of Procurement and Supply (CIPS) program, The Prevention, Preparedness and Consequence Management of Terrorism and other Security-Related Risks. The program was announced in 2007 by a European Commission–Directorate-General Justice, Freedom and Security in compliance with Council Decision No. 2007/124/EC [4]. The project was designed to ensure the physical protection of crucial facilities for the production, transmission, and distribution of energy systems. Within the project, the PPS standards [19] specify relevant levels of individual protective measures needed to prevent threats from terrorism and criminal activity.

Another project, Critical Infrastructure Protection in Energy Sector (CIPnES), was implemented in 2009–2011. This project, which was also part of the CIPS program, was interlinked with the previous project, where PPS standards for energy, gas, and oil industries had been created [21].

Alternative approach: The subject can select from a finite number of optional solutions in which different technical, organizational, or mode measures are combined [20]. Currently, the alternative approach is considered more efficient; it is based on the principle that many technical measures are needed so that the intruder can be detected and caught before reaching the target. The approach is based on mathematical and statistical methods, which,

through measurable input and output parameters of efficiency, reliability, or effectiveness, can verify the physical protection system. In this case, it is possible to verify whether the designed protective measures are sufficient.

Variable approach: The subject must adopt measures that are part of a PPS, which consider the breaking resistance of mechanical barrier measures, the response times of the intervention unit, and the detection probability of alarm systems [20].

The VEGA 1/0640/10 project of the National Grant Agency was undertaken in 2010 to model physical protection systems. It produced a methodology and a simple software tool designed to quantitatively assess the efficiency of an object's protection system.

Designed base models were published in an article in the *Journal of Homeland Security and Emergency Management* [20]. The basic difference between them is the approach to the intruder's decision making during the attack path choice (decision certainty and uncertainty) and the method for defining input parameters, which are considered either constant variables (deterministic modeling) or random variables defined by the appropriate probability distribution (stochastic modeling). The individual solutions differ predominantly in the:

- Interpretation of output parameters;
- Method for loading input values of parameters (predefined input values, input values loaded by the evaluator);
- Approach considering random effects (deterministic or stochastic);
- Intruder's decision method (certainty or uncertainty);
- Method for specifying the intruder's route (e.g., adversary sequence diagram or precise trajectory);
- Anticipated type of attack (destruction, damage, or theft);
- Modeling method of the guarded area (2D or 3D visualization, input matrix);
- Method for using sensitivity analysis.

The mathematical models used were the previously cited software tools (SAVI/ASSESS, Sprut, Vega-2, Analyser SFZ, SAPE, etc.) [14,22–26], the analysis of which showed both strong and weak points. When applied to critical infrastructure elements or within the EU member states' environment, these tools have certain disadvantages [20]:

- They were designed to protect specific materials and non-commercial facilities, not critical infrastructure elements with different modes of operation;
- They allow modeling and further simulating scenarios only of anthropogenic threats;
- They do not account for the level of protection in multilevel facilities to be assessed;
- They do not account for the level of protection in line facilities and elements to be assessed;
- They lack the modeling of a direct physical confrontation between the intruder and the intervention unit.

After the end of the programming period of the program Prevention, Preparedness and Consequence Management of Terrorism and Other Security-Related Risks (2007–2013), research in this area was redirected to the research scheme H2020 Secure Societies, which was understood in a broader context of improving the resilience of critical infrastructure. Examples include projects such as IMPROVER improved risk evaluation, and the implementation of resilience concepts in critical infrastructure: Realising European ReSiliencE for CritIcaL INfraStructure (RESILENS) and Advanced surveillance system for the protection of urban soft targets and urban critical infrastructures or Strategic, tactical, operational protection of water infrastructure against cyber-physical threats (SURVEIRON). Research in the field of critical infrastructure resilience was also conducted within the framework of national grant schemes of EU Member States. Project Dynamic Resilience Evaluation of Interrelated Critical Infrastructure Subsystems, supported by the Ministry of the Interior of the CR, focused on the dynamic evaluation of a correlation in significant European sectors (energy, transport, and ICT) and their components.

Game theory has been widely applied in computer models, as its principle is easily algorithmic. This approach has found many applications in cybersecurity [27]. In security management, it can model different conflicts, for example, modeling mixed threats and as a tool for warfare [28]. Another example is modeling the response to the disclosure of additional security measures, for example, with new technologies used in airport controls and the response to it by terrorists [29]. This approach can also be found in heuristic algorithms [30].

Game theory was constructed to find a solution to two-sided conflict; therefore, it is also excellent for modeling face-to-face clashes in the physical protection of critical infrastructure elements, as shown in this article. However, in security management, it is mainly used to model technical equipment [31,32]. Insider threat was also analyzed using game theory [33].

From a mathematical point of view, it would be interesting to implement the f adaptive game approach presented in [34] for clashes between intruders and defenders. Incorporating existing studies on human behavior, people's real-life choices [35], and adaptation of game theory to real data [36] is also very promising. However, for the physical protection of critical infrastructure, this approach has not been frequently used.

Game theory is a mathematical technique that can be successfully combined with other mathematical tools such as Bayesian probability [37], modeling of measurement accuracy [38], Saaty method [39], and many others, showing that game theory can be applied to modern models.

The detection of an intruder at the protected object is only the first step in the elimination of a potential threat. After detection, the intruder must be apprehended and escorted out of the protected area. This process depends not only on the intruder's skills and equipment but also on the physical protection system.

The face-to-face interaction (confrontation) can be the weak point of a physical protection system because it depends on human decision making and the qualities of the defender. Obviously, the response (protection) strategy is usually outside of the security manager's interest and depends on the defender's decision and skills. The defender's choices, as a member of a security service, may not be fully consistent with the security measures implemented by security managers.

Therefore, physical clashes between intruders and defenders are weak points of physical protection systems. The need to identify and validate the necessary and sufficient level of protection during the clash can be challenging for the defender.

To address this dilemma of the defender, basic strategies for selected types of intrusion scenarios should be planned in advance. These scenarios can be modeled with software tools as a game theory problem. Simulations of clash scenarios can help security managers to determine the optimal intrusion response scenario, identify essential skills that the defender should have, clarify gaps in an intuitive approach, and help to identify any impediments to the removal of the threat.

This approach was used in the project for Nuclear Regulatory Authority of the Slovak Republic: Integrated mathematical and computer system for determining the probability of correct detection and successful elimination of the intruder of the integrity of the physical and object security of the nuclear facility by the forces and means of physical protection of the nuclear facility II, stage 2015. Some of the results were published in [40–42].

In this article, we provide a discussion of the usage of the game theory approach for software tools such as SATANO to apply this mathematical method for a physical protection system design. We aspired to help members of security services to enrich their decision making with analytical tools.

## 3. Results

### 3.1. Security Assessment of Terrorist Attack in a Network of Objects (SATANO)

The Security Assessment of Terrorist Attack in a Network of Objects (SATANO) is a new software tool enabling the quantitative assessment of the level of PPSs for critical

infrastructure elements using various 2D map data, as shown in Figure 1. This software was created as part of the Critical Infrastructure Protection Against Chemical Attack (CI-PAC) project (HOME/2013/CIPS/AG/4000005073), undertaken between 2014 and 2016.

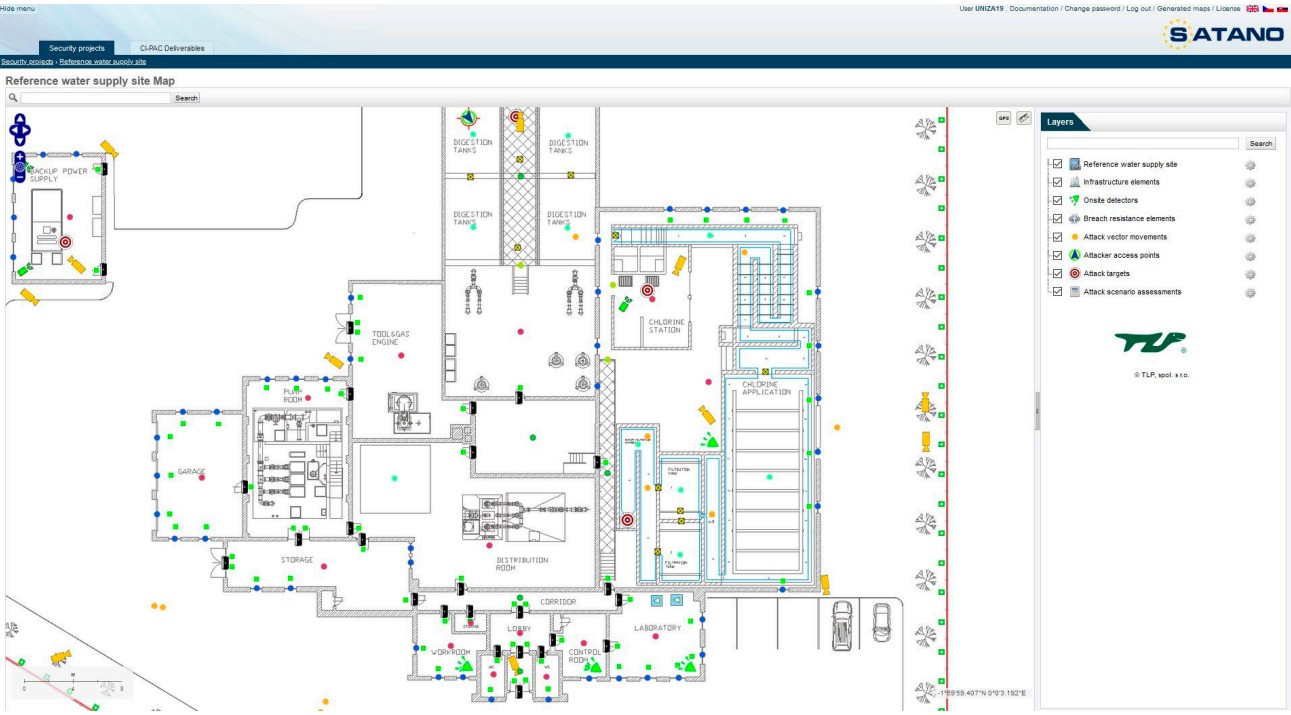

**Figure 1.** Graphical user interface of the SATANO software tool version 2.0.

The SATANO software tool uses the variable approach because it is based on the premise that the intervention unit must use as many protective measures as possible to detect and detain the intruder before they reach the target and damage or destroy the protected interest (the critical infrastructure element). The tool integrates the pessimistic (i.e., deterministic) model, which fully excludes any incidental effects that might occur during the attempt to penetrate the guarded area. The intruder is assumed to have all the information about the protected interest (they decide with certainty, i.e., they know the critical path) and know how to reach it. The total time begins when the alarm system signals that it has detected the intruder and covers the time required to break all barriers, including the sum of the shortest time for the movement between/among barriers.

In terms of making it possible to model the system of physical protection using various map data on a relevant scale (Figure 2), this tool, unlike other software tools (e.g., SAVI), is suitable for any tier building or line construction (e.g., airports, administration buildings, oil pipelines, and water supply sites).

The main innovative benefit of this newly created software tool is the possibility of creating more sophisticated attack scenarios than just an attack by a person using various types of tools, which can represent an attack vector. This tool can be used to simulate various intrusion scenarios.

In terms of physical protection, an attack vector can be an entity that has the potential to cause a negative effect because of its properties (physical and chemical) and abilities (knowledge, skill, and experience) [43,44]. In other words, the attack vector is the environment-determined procedure or method through which the vector (entity) accomplishes the attack within a particular space, direction, and time, as shown in Figure 3.

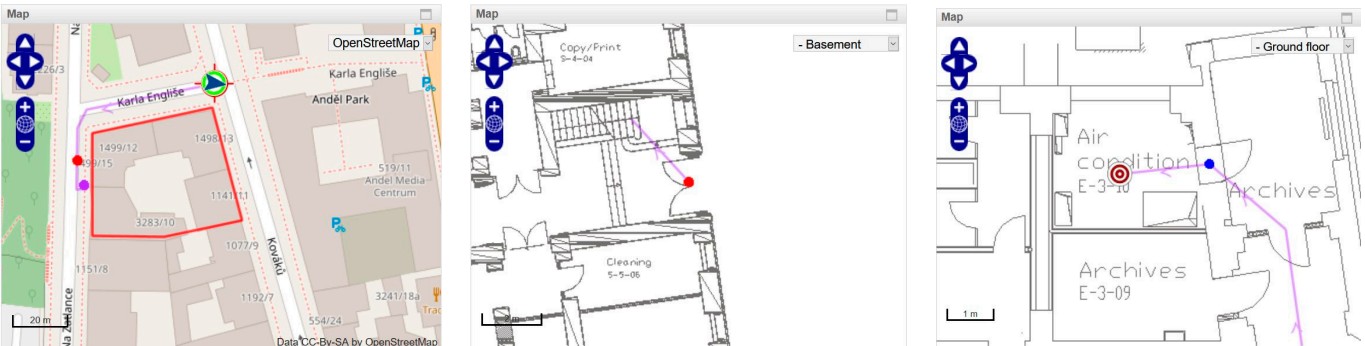

**Figure 2.** Modeling the PPS (SATANO software tool version 2.0).

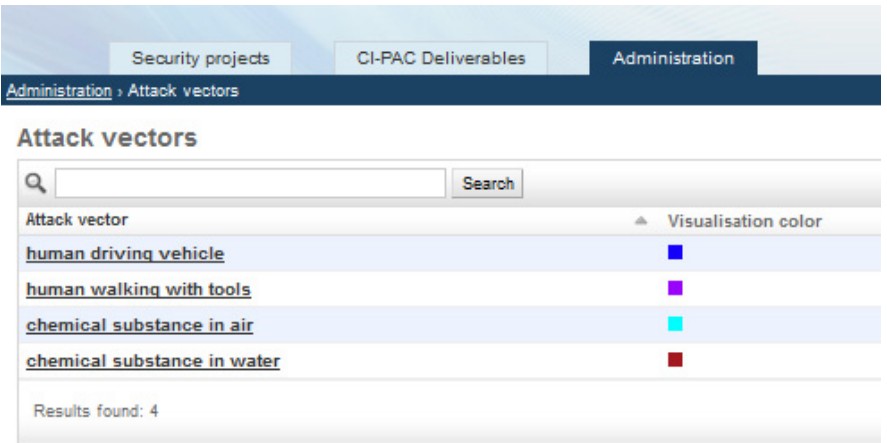

**Figure 3.** Predefined attack vectors (SATANO software tool version 2.0).

Current tools, except for SATANO, cannot model more complicated scenarios (based on more than one attack vector), such as those in which an intruder, having overcome barriers, releases a chemical agent into a ventilation or piping system.

The attack scenario within the SATANO tool is a description of 1 to N steps of the attack vector; the attack vector gradually moves from the access point to the target point, and it gradually reaches from 0 up to N-1 partial attack targets. The attack vector can optionally transform into a different attack vector in every partial attack target, as shown in Figures 4 and 5. Figures 4 and 5 show the creation of a scenario combining two attack vectors. The first part of the scenario simulates the attack on a chlorine station by a walking human using different tools (e.g., axe, hammer, drilling machine, etc.), and the second part of the scenario simulates the spread of a chemical substance in the drinking water supply for residents of a residential district, block D5.

The newly created software tool SATANO was first used in the process of assessing the level of protection of a particular element of critical infrastructure in the Slovak Republic. According to Act No. 45/2011 Coll., the Vodňany waterworks are by law designated a critical infrastructure element (CIE), as an engineering building situated within the borders of the Slovak Republic, for which disruption or destruction would have severe negative consequences on the quality of life and health of the state's population as well as on the environment.

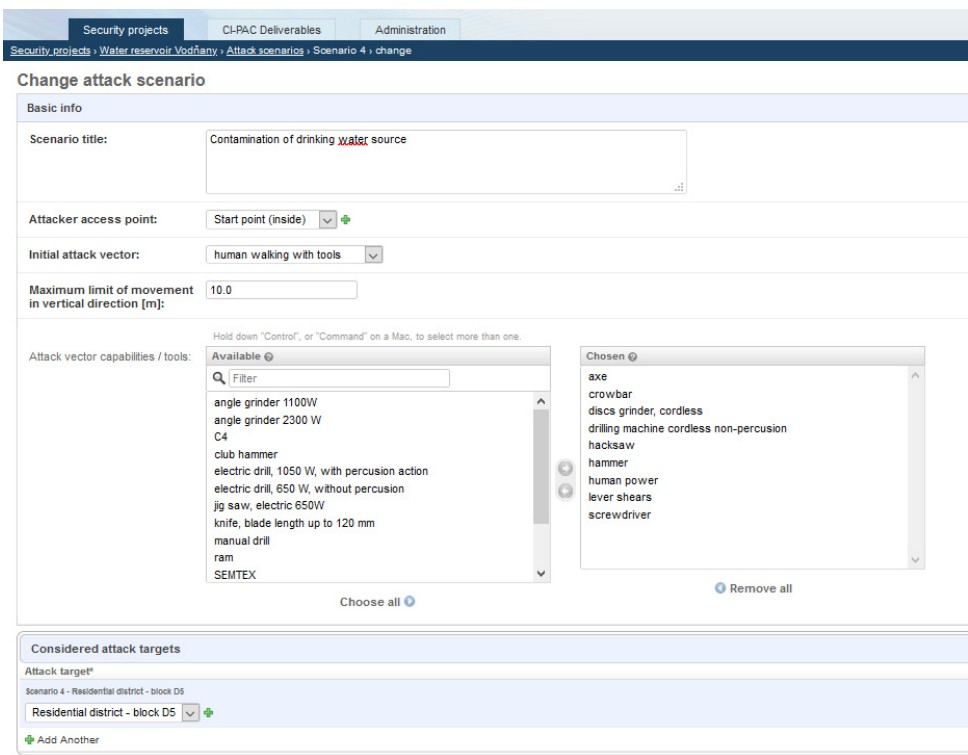

**Figure 4.** Choosing of attack vector (SATANO software tool version 2.0).

**Figure 5.** Change of attack scenario (SATANO software tool version 2.0).

According to Council Directive 2008/114/EC, it is necessary to consider relevant threat scenarios with the aim of reviewing weak points and the potential influence of disruption or destruction of the critical infrastructure. According to [3], risk analysis means consideration of relevant threat scenarios in order to assess the vulnerability and the potential impact of the disruption or destruction of critical infrastructure.

For waterworks, the specified relevant threat scenarios were subsequently simulated and evaluated using the SATANO software tool by calculating individual indicators and

critical paths, producing a graphical representation of the moment of detection as well as the timeline of the attack.

One of the scenarios is an attack by an external intruder with the aim of poisoning a large group of the population within the selected residential area. The intruder uses a powered hand glider to land in an area near a water chamber and subsequently overcomes the standard opening barriers using selected tools on their way to the chlorine station where they pour the chemical into the pumping device (Figure 6). In the scenario evaluation, the system was effective, with a 0.994 probability of interruption. Even though the task force does not eliminate the intruder, the water pipeline system is able to react in time and close the supply of drinking water to the D5 residential area [45].

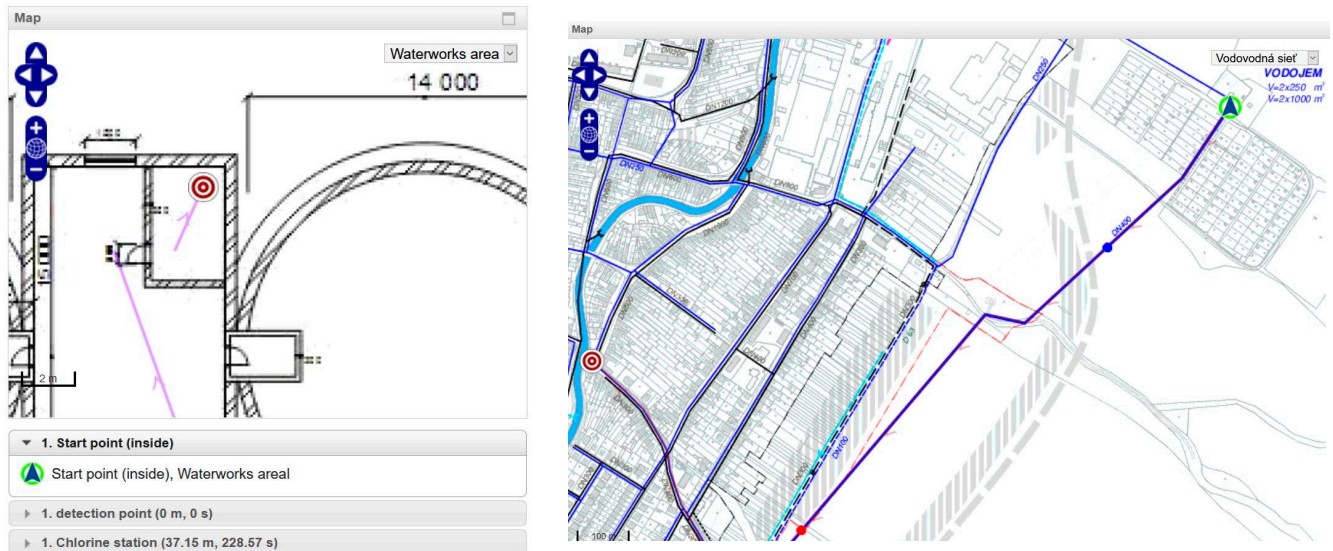

**Figure 6.** Graphical representation of the intruder's critical route in scenario (SATANO software tool version 2.0).

Figure 7 shows the timeline of the intruder's progress, which shows that the detection occurred immediately after their landing in the area of the waterworks and that the object was not secured by the task force in time. However, the chemical detector reacted to the contamination of the drinking water source and closed the system, which happened 480 s after the point of detection [45].

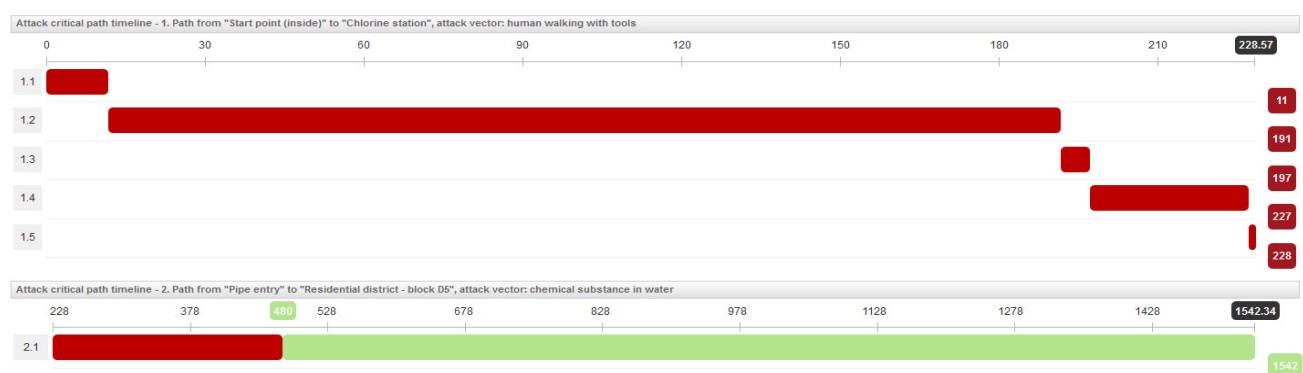

**Figure 7.** Timeline of the intruder's progress in scenario (red, time post-detection; green, additional time reserve for the task force) (SATANO software tool version 2.0).

SATANO, as a software tool with different input and output probabilistic and time parameters (e.g., probability of interruption), designed for the quantitative assessment of PPSs,

provides additional functionality. For example, SATANO has the potential to be combined with other approaches such as the Structure Analytic Hierarchy Approach (SAHA) [46].

In the future, a new mathematical model (Intrusion-Response Scenario) will be implemented using new output parameters, an optimal intrusion-response strategy, to objectively assess the level of PPSs from the perspective of the intruder and intervention unit conflict.

This variable can provide additional information about situations in the protected area.

### 3.2. Selection of Intrusion-Response Scenario

Software tools used in physical protection systems are immensely popular because of their ability to simulate situations, for which it would be resource-intense to prepare exercises or other measurements in real life. However, the physical clash of the defender and intruder is difficult to precisely simulate because of the wide variability in inputs and outputs.

Modeling this clash is an important part of examining the conditions of the application of a certain level of physical protection. This is because it is the human factor that is the riskiest aspect of the physical protection system. Barriers, detection systems, and intruder alarms can slow, discourage, or even stop intruders, but it is the security services that take the intruder to the criminal authorities that make the physical protection system truly effective. If this factor is missing, it may lead to an increase in efforts to intrude in the building, as the intruders would not be punished.

In addition, preparing strategies to respond to the disruption of critical infrastructure elements is an important part of increasing the mental preparedness of defenders. The introduction and repetition of activities to eliminate the intruder (stopping, detaining, disabling further action, and removing the intruder from the protected area) improve their effectiveness, as the defender does not find themselves in a completely new situation and does not need to look for new solutions. For interventions for those who train responses for certain situations, the results of such a simulation can provide more knowledge of the problem and better orientation in which the strategy is beneficial.

A practical method to determine an appropriate response is to model a confrontation entailing the physical defense of a protected interest in the terms of the game theory introduced by Neumann [47]. The generality of this model allows quick variation in the situations, skills, and qualities of the confrontation.

The theory presented in the next subsection explains the used approach, game theory, in terms of the selection of intrusion-response strategy and presents the mathematical background to determine the appropriate defender behavior.

### 3.3. Calculation of Intrusion-Response Strategy

The problem of a clash between an intruder and a defender can be perceived as a conflict (game) with two sides, where each side chooses how it will behave (strategy), which places the problem into game theory from the mathematical (modeling) point of view. The intruder and defender can be called players.

In this specific case of a non-cooperative game, victory for one player means a loss for the other, and vice versa (zero-sum game). Before an attack, it is not clear how the intruder will behave, or which features they will use. However, the possible strategies between which the intruder will choose are known. This set can be known through the experience with the object. If the protected object is a hydroelectric power plant, the intruder is likely to proceed differently than with a nuclear power plant. In addition, intrusion strategies can be estimated based on historical experience, knowledge of the environment, and the demographic distribution around the protected interest. As such, the set of attack scenarios (now meaning the behavior of the intruder, not the path of attack), called intruder's strategies, and the defender's behavior, called defender's strategies, can be found.

The strategies of both the intruder and defender can be represented as a player's quality function. This function can be a composition of various parameters used to model

the performance of the intruder or defender in a selected situation. Some parameters can be measured and others estimated, while the relationship between them should be captured in the quality function. This procedure was chosen in a previously mentioned project for the Nuclear Regulatory Authority of the Slovak Republic, for which partial results can be found in [40–42].

In this step, it is necessary to construct a list of all rational expected behavior that the intruder may choose during the physical clash. For example, if the intrusion is misbehavior of intruders, the defender will end the clash with a verbal call to leave. The possible intrusion strategies can be determined by analyzing the types of intrusions. There are few aspects that help with modeling intrusion strategies. The intruder will proceed with the physical clash depending on if they are prepared, if they planned the intrusion, how trained they are, and if they mean to harm the defender in some way or not. This can help to scale the levels of a physical clash between an intruder and the defender and create the intrusion strategies, i.e., the intruder's quality function.

Each situation where the intruder acts according to some strategy and the defender follows one of their strategies, a text intrusion-response strategy, is described by a combination of intrusion and response schemes called the payoff matrix.

In game theory, this situation can be described as payoff matrix $M$, where the elements of the payoff matrix depend on the ratio of protection quality function to the intruder's quality function $m_{ij}$. The payoff matrix $M$ demonstrates the efficiency of the response strategies against the intruder's strategies:

$$M = \begin{pmatrix} m_{11} & \dots & m_{1c} \\ \dots & \dots & \dots \\ m_{r1} & \dots & m_{rc} \end{pmatrix} \tag{1}$$

where $m_{ij}$ denotes the result of the clash between the intruder using the $j$th strategy and the $i$th strategy of response. These strategies can differ, and the matrix does not have to be a square matrix.

However, because of the complexity of the building intruder's quality function process, the determination of the intruder's quality function as a set of parameters and their inner dependencies for individual strategies may be omitted. Then, the game matrix is created directly so that the individual elements of the matrix represent an estimate of which side will win the clash and in what proportion. For example, the first intrusion strategy is half as strong as the first defender strategy.

The numbers in the payoff matrix can be set using experts' estimations if it is not possible to analyze each intruder or defender quality function.

In the case where lists of both intrusion and response strategies are set, it is natural to assume that each player wants to maximize their profit while minimizing their loss. This assumption means that the players are rational and non-cooperative. Neumann created this assumption [47] when the game theory was presented.

Then, the optimal choice of strategies for both rational players is given by [48]:

$$\max_{i \leq r} \min_{j \leq c} m_{ij} \leq \min_{j \leq c} \max_{i \leq r} m_{ij} \tag{2}$$

where $m_{ij}$ denotes the result of the clash between the intruder using the $j$th strategy and the $i$th strategy of the response following the same notation as in the payoff matrix.

Since the game is zero-sum, the optimum ($s$) is an element of the pay-off matrix, where the equivalence $s = \max_{i \leq r} \min_{j \leq c} m_{ij} = \min_{j \leq c} \max_{i \leq r} m_{ij}$ holds. This point is called the saddle point [48].

The optimal intrusion-response strategy is created to effectively respond to the behavior of the intruder, whereas the intruder seeks to maximize the effect of their action.

In this clash between two opposing interests, the goal is to find the element of the payoff matrix called the saddle point, which is found by assuming that problem elimination is a normal-form game between two rational players in one move. Rationality, in this case,

means that both the intruder and defender can calculate the payoff matrix and want to maximize their own benefit while reducing the opponent's benefit [47]. This means that the response strategy is built on the knowledge that the intruder assumes that there will be some response against their actions and tries to find how to benefit as much as possible from the situation. In addition, the response strategy tries to lower the negative effects of intrusion to the maximum extent possible.

Under this assumption, the solution, if it exists, is easy to find and is the saddle point of the payoff matrix M. This point determines the optimal intrusion-response strategy. Thus, if the strategy of the participant differs from the saddle point (optimal intrusion-response strategy), their payoff from the game is reduced, meaning that finding the optimal intrusion-response strategy produces the response strategy that minimizes the intrusion effect for all expected intrusion scenarios.

If a solution does not exist, the saddle point or optimal intrusion-response strategy does not exist. Nonexistence of the saddle point can also lead to another type of game in game theory. It means that another approach should be adopted to determine how to behave. In this case, it may be effective for physical protection system designers or security service members to use information collected in another way. For example, the intrusion strategies can be compared one by one to all found defender strategies, as was the case in the payoff matrix; for each intrusion strategy, the best response strategy should be separately found. This approach will lead to a list of defender strategies for certain types of intruder strategies. This list would complete a software solution such as SATANO and enhance the assessment of the physical protection of an examined object.

Even if the model describes the approach to determinate the best-expected response strategy for a defender in some physical protection system, it is still convenient to investigate the effectiveness of the intrusion-response strategies. Understanding the intrusion-response strategy effectiveness provides more information about the level of the physical protection system, its preparedness, and its weak points.

The model seems suitable; the condition of rationality cannot be guaranteed as the rationality of both participants can be lowered by many factors. To increase the reality of the model, it is necessary to introduce the irrationality measure [41]:

$$irr_{ij} = \left| s - m_{ij} \right| \tag{3}$$

where $s$ is the saddle point and $m_{ij}$ is the element of the payoff matrix.

Irrationality allows the intruder and defender to choose a strategy different from the optimal strategy. It is used to calculate the probability of successful response to the $i$th intrusion quality function using the selected $j$th response quality function.

It is advantageous to use the irrationality measure especially when it is not possible to set specific intruder and defender strategies. It can even be used when defenders are not sufficiently trained in a large range of defense strategies or the object has a large flow of defenders. This measure replaces the uninformed decisions made by the defender if they did not use this model. However, the advantage is that the determination of this rate is based on objective rational considerations and modeling. To clarify this point, an example is provided below.

Notably, these results are only applicable for the response strategies noted at the left side of the payoff matrix *M*. This means that the defender is looking for the optimal strategy and the defender is looking for irrational behavior. Nevertheless, the intruder can also choose any strategy other than the optimal strategy, which can lead to lowering the payoff for the defender from the game.

For example, let us take the following matrix of six different intruder strategies and five response strategies. The numbers in the matrix are set as an estimation of the ratio between intruders and response strategies, meaning the intrusion IN(1) is half as strong as the response strategy RE(1). The strategies can be established as a function of some parameters such as strength, knowledge of an environment, and psychological components, as reported in [41]. Another method to set the intrusion-response matrix is estimating the ratio between

the assumed strategies from their description, as was conducted here. Additionally, there are many other possibilities for estimating parameters in security models [37,38,49].

The following Table 1. represents an example of Intrusion-Response matrix. The intrusion strategy IN (1) represents, for example, harmless adolescent vandalism; IN (2) can be small theft, and so on, up to terrorism acts. RE (1) can be a rebuke and RE (5) can represent a physical response. The set of strategies depends on the protected object.

**Table 1.** Saddle point in the intrusion-response matrix.

| Strategies | IN (1) | IN (2) | IN (3) | IN (4) | IN (5) | IN (6) | Min |
|---|---|---|---|---|---|---|---|
| RE (1) | 0.5 | 0.6 | 0.1 | 0.5 | 0.7 | 0.5 | 0.1 |
| RE (2) | 0.2 | 0.3 | 0.2 | 0.2 | 0.6 | 0.4 | 0.2 |
| RE (3) | 0.7 | 0.5 | **0.4** | 0.7 | 0.4 | 0.7 | 0.4 |
| RE (4) | 0.7 | 0.4 | 0.1 | 0.1 | 0.2 | 0.8 | 0.1 |
| RE (5) | 0.8 | 0.5 | 0.3 | 0.8 | 0.6 | 0.3 | 0.3 |
| max | 0.8 | 0.6 | 0.4 | 0.8 | 0.7 | 0.8 | 0.4 |

The Irrationality measure can be calculated from the Table 1 by rule for irrational measure defined in the Equation (3).

The saddle point (0.4) is found based on Neumann's theory explained above (Equation (2)). It means that the response strategy is built on the knowledge that the intruder assumes that there will be some response against their actions and tries to find a way to gain as much from the situation as possible. Conversely, the response strategy tries to lower the negative effect of the intrusion.

The zeroes in Table 2 indicate non-irrational decisions; in other words, rational decisions. Notably, these decisions are only applicable to response strategies (always at the left side), since they are obtained from the matrix *M*. This means that the responder is choosing an adequate strategy. If the responder is not fully informed, they might also choose response strategies RE (2) and RE (4). Nevertheless, the intruder may choose a strategy other than optimal strategy as well, which can lead to lowering the total gain from the game; in other words, response strategies RE (2) and RE (4) would be less effective against all other choices of the intruder. RE (2) and RE (4) strategies allow the intruder to obtain a higher score (0.6, 0.7 and 0.8) than the initial 0.4.

**Table 2.** Irrationality measure in the intrusion-response matrix.

| Strategies | IN (1) | IN (2) | IN (3) | IN (4) | IN (5) | IN (6) |
|---|---|---|---|---|---|---|
| RE (1) | 0.1 | 0.2 | 0.3 | 0.1 | 0.3 | 0.1 |
| RE (2) | 0.2 | 0.1 | 0.2 | 0.2 | 0.2 | 0 |
| RE (3) | 0.3 | 0.1 | 0 | 0.3 | 0 | 0.3 |
| RE (4) | 0.3 | 0 | 0.3 | 0.3 | 0.2 | 0.4 |
| RE (5) | 0.4 | 0.1 | 0.1 | 0.4 | 0.2 | 0.1 |

Finding the irrationality matrix provides additional information about the response choices for an already-fixed intrusion scenario and also helps to analyze the results of the simulated behavior of response strategies. During the simulations, it is easy to change the input values of the player's quality functions and thus monitor for which values there are large and small deviations from the rational solution. By comparing the input values whose rationality is low with the strategies that have been chosen, it is possible to identify the conditions for the choice of strategies, or situations in which an irrational choice may occur.

In the project for the Nuclear Regulatory Authority of the Slovak Republic, the quality functions of the intruder and defender were built upon the same parameters for each quality function. This allowed the simulation of irrationality to be used for finding crucial parameters for the defender's response. In addition, this information has been used in training and preparedness exercises.

## 4. Discussion

In their research at our home institution, we have long been striving to develop and enhance a new, more complex approach to SATANO as a software tool for modeling and simulating various attacks on critical infrastructure elements and for the quantitative assessment of the physical protection system.

The new software tool SATANO and the theoretical intrusion-response strategy model provide significant help for designers while preparing physical protection systems for critical infrastructure elements and determining their level of protection. In contrast with its predecessors, SATANO has many distinct advantages, which can facilitate its broad application within the sphere of the strategic protection of state objects.

Although SATANO overcomes several disadvantages of the existing software tools and expands their use for other attack vector types, it still cannot model clashes between an intruder and a defender. However, there are several tools that use real-time simulation of the confrontation (e.g., Joint Combat and Tactical Simulation). Such simulation tools focus on the visualization of force-on-force exercises and omit face-to-face clashes. This approach to a confrontation involving whole-unit simulation is essentially only experimental, occurs in real-time, and provides virtual verification of the reaction of the intervention unit against intruder attack in certain default scenarios. In tools such as Joint Combat and Tactical Simulation, it is not possible to change the input values of the face-to-face clash scenario and, thus, optimal intervention unit response cannot be sought within the created model of the PPS.

The model can describe and simulate different clashes between an intruder and a defender in the critical infrastructure element. This model was presented in this paper as an intrusion-response strategy model. It allows users to describe the chances of intrusion strategies and to find adequate response strategies. Since it is a mathematical model, it can be easily implemented in a software tool, which will run simulations on it.

The intrusion-response model describes the adequate response strategies for given intrusion types. After comparison of expected and commonly used response strategies, it allows the determination of the level of physical protection systems for different intrusions.

The model identifies the optimal strategy that can be used in a situation where the defender has no other information other than an intruder being detected in the protected object. Moreover, it identifies other suitable response strategies without all the necessary information by using the irrationality measure.

The strategies of the participants in this model depend on their skills and characteristics. Hence, the inputs of this model are parameters that describe them. Since it is not possible to measure all the values of a potential intruder's parameters, the values can be varied in simulations or estimated by experts.

The main advantage of the proposed method is its flexibility. Since the intrusion-response strategy model is based on general mathematical theory, it can be applied to a wide range of problems associated with training, planning, and evaluating response strategies. These strategies can be flexibly changed so that they respond to changes in the social environment, moods in society, and changes in the equipment of intruders, their tactics, and the motivation of attacks. In [41], strategies were determined by a combination of measurements and estimations as part of one of the tasks of creating the right response to selected attack strategies. It was possible to create simulations of the actual response of the defender to the intruder's actions.

SATANO with the intrusion-response model provides an effective approach to designing physical protection systems with adequately prepared defenders who know strong and weak points.

This is an alternative approach to the problem of the evaluator's personal influence in physical protection systems. This approach is based on mathematical models and software solutions, here represented by SATANO and the intrusion-response strategy model.

SATANO was designed to quantitatively assess the level of PPSs for critical infrastructure elements with graphical interpretation and visualization in 2D maps. This model, with

its tools, is suitable for any tiered building or line construction (e.g., airports, administration buildings, oil pipelines, and water supply sites).

The intrusion-response strategy model is suitable for complementing existing tools of SATANO to help members of security services find the proper response strategy to different intrusions, and PPS designers to understand the effect of PPS design on response strategies.

In this manner, each nation can individually address the interpretation of SATANO and the intrusion-response strategy model results, which can help to identify and assess risks, and select and prioritize countermeasures and procedures for elements with strategic importance in its legislation.

The new software tools can increase the level of the physical protection system. This enables better control of the protection of critical infrastructure that meets the essential needs of citizens. Therefore, due to modeling of CI physical protection systems of and simulations of possible attacks is possible to ensure long-term, sustainable, and, mainly, more objective security of citizens, cities and states.

**Author Contributions:** Conceptualization, T.L.; methodology, K.K.; software SATANO, T.L.; validation, T.L.; formal analysis, K.K.; investigation, L.S.; resources, T.L., K.K., and L.S.; game theory solution, L.S.; writing—original draft preparation, T.L., L.S.; writing—review and editing, K.K.; visualization, K.K.; supervision, T.L.; project administration, K.K.; funding acquisition, T.L., K.K. All authors have read and agreed to the published version of the manuscript.

**Funding:** This research received no external funding.

**Institutional Review Board Statement:** Not applicable.

**Informed Consent Statement:** Not applicable.

**Data Availability Statement:** Not applicable.

**Acknowledgments:** This research was co-funded by the EC, DG Home Affairs Prevention, Preparedness and Consequence Management of Terrorism and Other Security-related Risks Programme of the EU, addressed in 2014–2016, HOME/2013/CIPS/AG/4000005073. This research was co-funded by the Ministry of Education, Science, Research, and Sport of the Slovak Republic in 2018–2020, VEGA 1/0628/18 "Minimizing the level of experts' estimations subjectivity in safety practice using quantitative and qualitative methods".

**Conflicts of Interest:** The authors declare no conflict of interest.

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
