# Peer review of "Modeling and Simulation as Tools to Increase the Protection of Critical Infrastructure and the Sustainability of the Provision of Essential Needs of Citizens"

_sustainability, doi:10.3390/su13115898_

Round 1

Reviewer 1 Report

The paper addresses the problem of finding an appropriate approach for establishing appropriate physical protection systems for elements in critical infrastructure on the national level which provide and maintain essential services to citizens. This paper proposes a quantitative approach to design physical protection systems by application of a new software solution called SATANO for which authors propose mathematical extension based on the Game theory. Results are divided into two complex parts. In the first part, the software SATANO is described in detail and very understandable way. The figure 3 in this section should be divided into two separated pictures with separated descriptions. Since it is hard to read the text inside of it. Authors should provide examples where this SATANO software was used or is planned to be used. The second part deals with a mathematical model designed to find an optimal intrusion-response strategy. This strategy can be identified with a lack of information that gives the defender an advantage when deciding in security situations. This section uses classical Neumann theory efficiently described for usage in security. On the other hand, Equation 3 seems new and should be described in more detail. The authors should better explain how this irrationality measure can be used. Besides these completions I find this research article innovative. 

Author Response

Dear Editor-in-Chief and Reviewer,

Thank you for sending valuable comments and the opportunity to improve the quality and clarity of our article. We would also like to thank the reviewers for their valuable comments that we have tried to elaborate in full. The authors 'response to reviewers' comments you can find below.

Sincerely,

Authors

Reviewer #1:

  1. The figure 3 in this section should be divided into two separated pictures with separated descriptions. Since it is hard to read the text inside of it.

We agree, the Figure 3 was divided into two separate images with a short description. [Lines: 1442 - 1522]. 

  1. Authors should provide examples where this SATANO software was used or is planned to be used.

Thank you very much for your feedback. To the paper was add example of application SATANO software in the assessing process of protection level of waterworks Vodňany, which is by Slovak law designated as an element of critical infrastructure in Slovak republic. [Lines: 1563-1586]. 

  1. On the other hand, Equation 3 seems new and should be described in more detail. The authors should better explain how this irrationality measure can be used.

Thank you for this comment. The Equation 3 indeed needed better explanation. Its usage is now commented above it. Lines: [2240-2246]. 

And it is mentioned also in the example. Lines: [2504-2533]. 

Reviewer 2 Report

The present manuscript introduces a software simulation model for assessing the unauthorized intrusion and introduces potential applications as a decision support solution.

The introduction presents an overview of the domain and some related software. The first section that describes cities and SDG appears to be lacking coherence to the follow-ons. Some in depth analysis of existing literature could be used especially linked to the objectives of the paper. Also introduced projects could be updated as many of them they appear to be completed more than a decade. Presented software are highly relevant. 

  The mathematical representation of the SATANO, based on game theory, allows to model response scenarios of attackers and this has been reflected in the text. Exemplary presentation of intruder and defender strategies would be beneficial for future readers.

A explanation of the numbers in the payoff matrix, and especially how to derive them is missing. Probably this is a highly complex process, but this issue leaves some rather “incompleteness” of the work.

Author Response

Dear Editor-in-Chief and Reviewer,

Thank you for sending valuable comments and the opportunity to improve the quality and clarity of our article. We would also like to thank the reviewers for their valuable comments that we have tried to elaborate in full. The authors 'response to reviewers' comments you can find below.

Sincerely,

Authors

Reviewer #2:

  1. The first section that describes cities and SDG (Sustainable Development Goals) appears to be lacking coherence to the follows.

Thank you very much for your feedback. In the introduction, we highlighted the connection between global naturogenic and local anthropogenic threats with impact to states, cities and their strategic infrastructure. [Lines: 79-175]. 

  1. Some in depth analysis of existing literature could be used especially linked to the objectives of the paper.

Thank you for the observation. Additional literary sources have been added to complement the text of the article. Lines: [1076-1077, 1088-1091, 1442, 2510-2512]. 

  1. Also introduced projects could be updated as many of them they appear to be completed more than a decade.

Thank you very much for comment. After the end of the programming period of the program "Prevention, Preparedness and Consequence Management of Terrorism and Other Security Related Risks (2007-2013) research in this area has been redirected to the research scheme H2020 Secure Societies and was understood in a broader context of critical infrastructures resilience improving. We added some specific projects. [Lines: 884-1170]. 

  1. Exemplary presentation of intruder and defender strategies would be beneficial for future readers. Thank you very much for pointing the problematics of setting the strategies out. There were used two approaches so far. One is to take strategies as the functions and to measure and estimate their parameters as it was done in project for Nuclear Energy Research Institute - VUJE, a.s. The other is to estimate the ration between intrusion strategy and response strategy as it can be found in the example. [Lines: 2505-2511, 2833-2838]. 

  1. A explanation of the numbers in the payoff matrix, and especially how to derive them is missing.

Thank you for this comment. The example of finding an optimal Intrusion-Response strategy was added to the Result section to present the usage of this part of the Game theory in protection of physical objects. [Lines: 2504-2533]. 

Reviewer 3 Report

In the manuscript, the authors proposed a new software tool "SATANO" and a corresponding theoretical Intrusion-response strategy, which model and simulate various attack vectors on critical infrastructure elements and quantitative assessment level of the physical protection system. To enhance the readability of this manuscript, it is suggested:  

  1. The title and abstract in the system and the manuscript is not identical. Please make sure the right title & abstract are recorded. 
  2. The advantage of the proposed intrusion-response strategy needs to be highlighted, particularly in the conclusion and abstract sections. 
  3. Following the last comment, in the materials and methods section, the authors mentioned the mathematical models used previously. It is suggested that the authors extend this part a little bit, briefly elaborating these models used in SAVI/AS-SESS, Sprut, Vega-2, Analyser SFZ, SAPE, etc.
  4. A real implementation of the proposed intrusion-response strategy model should be included in the results section, for an explicit demonstration of the model capabilities. 
  5. Proper proofreading and English editing are suggested. 

Author Response

Dear Editor-in-Chief and Reviewer,

Thank you for sending valuable comments and the opportunity to improve the quality and clarity of our article. We would also like to thank the reviewers for their valuable comments that we have tried to elaborate in full. The authors 'response to reviewers' comments you can find below.

Sincerely,

Authors

Reviewer #3:

  1. The title and abstract in the system and the manuscript is not identical. Please make sure the right title & abstract are recorded.

Thank you very much for the warning. The title and abstract was updated in the system.

  1. The advantage of the proposed intrusion-response strategy needs to be highlighted, particularly in the conclusion and abstract sections.

Thank you for the advice. Based on it the advantages of the Intrusion- Response strategy model was added to abstract and discussion section.  [Lines: 63-65, 2831-2839]. 

  1. Following the last comment, in the materials and methods section, the authors mentioned the mathematical models used previously. It is suggested that the authors extend this part a little bit, briefly elaborating these models used in SAVI/AS-SESS, Sprut, Vega-2, Analyser SFZ, SAPE, etc.

We accept the comment. We have added a more detailed description of SW tools SAVI/ASSESS, Sprut, Vega-2, Analyser SFZ, SAPE, etc. [Lines: 408-425].

  1. A real implementation of the proposed intrusion-response strategy model should be included in the results section, for an explicit demonstration of the model capabilities.

Thank you for this comment. The example of finding an optimal Intrusion-Response strategy was added to the Result section to present the usage of this part of the Game theory in protection of physical objects. [Lines: 2504-2533]. 

  1. Proper proofreading and English editing are suggested.

The paper will be send to proofread.

Round 2

Reviewer 3 Report

Thank you for taking my suggestions into consideration. I recommend this paper be published. Good luck with your future research!